

# Forecasting agricultural drought: the Australian Agriculture Drought Indicators

Andrew Schepen[1], Andrew Bolt[2], Dorine Bruget[3], John Carter[3], Donald Gaydon[2], Mihir Gupta[4], Zvi Hochman[2,5], Neal Hughes[4], Chris Sharman[6], Peter Tan[4,] Peter Taylor[6]

[1]Commonwealth Scientific and Industrial Research Organisation, Dutton Park, QLD, Australia
[2]Commonwealth Scientific and Industrial Research Organisation, St Lucia, QLD, Australia
[3]Queensland Government Department of Environment, Tourism, Science and Innovation, Dutton Park, QLD, Australia
[4]Australian Bureau of Agricultural and Resource Economics, Canberra, ACT, Australia
[5]Agricultural Systems Consultant (zvi.hochman@gmail.com)
[6]Commonwealth Scientific and Industrial Research Organisation, Sandy Bay, TAS, Australia

*Correspondence to*: Andrew Schepen (andrew.schepen@csiro.au)

**Abstract.** Drought is a recurrent and significant driver of stress on agricultural enterprises in Australia. Historically, rainfall indices have been used to identify drought and inform government responses. However, rainfall indicators overlook important factors such as drought propagation and commodity prices. To address these shortcomings, AADI (Australian Agriculture Drought Indicators) was recently developed to monitor and forecast drought for upcoming seasons using biophysical and agro-economic models, including crop yields, pasture growth, and farm profit at ~ 5 km$^2$ resolution. Here, we evaluate the skill of drought indicator forecasts driven by the ACCESS-S2 dynamical global climate model over a hindcast period from 1990–2018. Analysis of the AADI hindcasts finds that antecedent landscape conditions significantly enhance predictive skill for crop yields, pasture growth, and farm profit across a financial year. As lead time shortens from 12 to 3 months, forecast confidence increases: median farm profit indicator skill rises from 43% at 12 months to 67% and 73% at 6 and 3 months, respectively, whilst median farm profit biases remain below 2% across all lead times, with high reliability indicating a well-calibrated ensemble, making the forecasts highly suitable for risk management and decision-making. Forecasts for wheat, sorghum, and pasture are also skilful and reliable in ensemble spread, although residual biases can occur (e.g., up to 20% for sorghum), which suggests further system refinements are needed. Analysis of historical events in both dry and wet conditions demonstrated the AADI system's ability to identify drought-impacted areas with increased confidence up to 6 months earlier than rainfall deficits.

## 1 Introduction

Drought is a recurrent and significant challenge in Australia, which affects water resources, agriculture and ecosystems (Van Dijk et al., 2013; Devanand et al., 2024; Holgate et al., 2020; Lindesay, 2005). Historically, government responses to drought





impacts in the agriculture sector have been informed by meteorological drought indicators such as rainfall deficits. However, a long history of practice has demonstrated that rainfall indicators are often poor proxies for agricultural and economic drought impacts (Hughes et al., 2022a; Das et al., 2023; Stagge et al., 2015; Wang et al., 2022). In the absence of accurate assessments

of agricultural impacts, government drought responses can be poorly directed, and overly reactive to media narratives (Rutledge-Prior and Beggs, 2021). Addressing these challenges requires not only monitoring of drought conditions, but also forecasting of drought evolution, including both onset and  recovery. (Das et al., 2023; Stagge et al., 2015; Wang et al., 2022). Whilst many drought warnings systems have been developed globally, most focus on meteorological indicators and emphasize monitoring over forecasting (Van Ginkel and Biradar, 2021). In Australia, tools like the AussieGRASS model (Carter et al.

2000) have long provided forecasts of agricultural indicators like pasture growth using analogues selected according to Southern Oscillation phase (https://www.longpaddock.qld.gov.au/AussieGRASS). In recent work, Bhardwaj et al. (2023) combined rainfall, soil moisture and evapotranspiration into a Principal Component Analysis (PCA) index, which they then paired with seasonal rainfall forecasts to develop a drought concern matrix. For example, dry antecedent conditions coupled with a high likelihood of low rainfall correspond to the highest level of drought concern. Whilst Bhardwaj et al. (2023)

demonstrated the use of seasonal forecasts in a drought early warning system, more work is needed to understand the regional economic impacts of agricultural drought across the combined cropping and grazing sectors, particularly as drought impacts can be modulated by external factors such as commodity prices.

The Australian Bureau of Agricultural and Resource Economics and Sciences (ABARES) has developed a statistical farm microsimulation model – *farmpredict* – which integrates climate data and farm survey data to predict consolidated farm profit

for a region (Hughes et al., 2019; Hughes et al., 2022b). An example of *farmpredict* output is provided in Figure 1 showing the expected relative farm profit for the financial year 2018-19, where the negative impact of the Tinderbox drought (Devanand et al., 2024) on farm profits can be seen over southeastern Australia. The question arises, then, whether *farmpredict* can be successfully driven by climate forecast ensembles to generate skillful forecasts of financial outcomes for farm businesses.




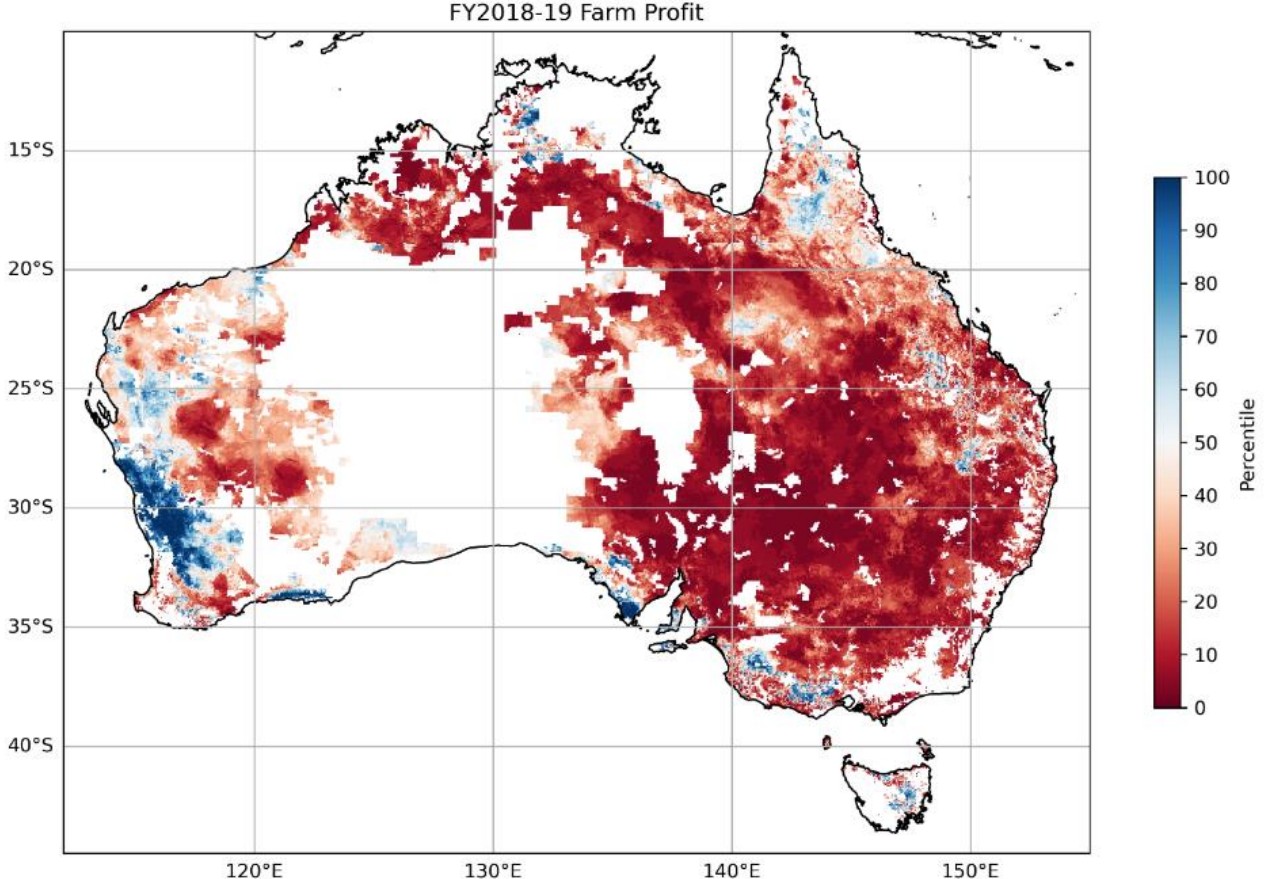

**Figure 1: Percentiles of farm profit for the 2018-19 financial year as output by the *farmpredict* model using a 1990-2018 baseline. The shaded regions are agricultural land and constitute the study area.**

Such an approach is not without precedent, as climate forecast ensembles have been used to generate seasonal outlooks of crop yields (e.g. Potgieter et al., 2022; Schepen et al., 2020b). Indeed, in addition to farm profit outlooks, drought analysts and policy planners can also be interested in the outlooks for winter and summer crops, or pasture availability for livestock.

Here, we connect climate forecasts from the Bureau of Meteorology's ACCESS-S2 seasonal model to *farmpredict*, crop models and a pasture model. Initially, climate forecast post-processing is needed to develop forecast ensembles that have the same characteristics as the observations used in the downstream models. New drought indicator forecasts are generated by forcing the downstream models with the post-processed climate ensembles on a 5km grid across Australia. Corresponding pseudo-observations of the target variables, which we call historical simulations, are generated by forcing the same models with observational weather data.

In this study, we assess the forecast performance of four agriculturally focused drought indicators: pasture growth, wheat yield (representing winter crops), sorghum yield (representing summer crops) and farm profit, that form the AADI (Australian Agriculture Drought Indicators) system (Hughes et al. 2024). Often, drought system performance is evaluated using threshold





or categorical forecasts (Madrigal et al., 2018; Sutanto et al., 2020; Li et al., 2023). Here, we verify forecasts using ensemble verification metrics and an ensemble of 51 members, which allows assessment across the range of thresholds and performance

for non-drought years as well as drought years. Forecast performance is investigated in terms of bias, accuracy (using the continuous ranked probability score), reliability (using probability integral transforms) and sharpness (using the interdecile range (IDR)). Additionally, we analyse the evolution of forecast for major historical events.

The paper is subsequently organised as follows. Sections 2 and 3 present the data and methods, including a summary of the climate, biophysical and *farmpredict* models, climate forecast downscaling, cross-validation and forecast verification. In

sections 4 and 5, the results are presented and discussed, respectively. Section 6 concludes the paper with the main findings. It is anticipated that the results of this study will support the operational AADI system by providing drought analysts with a level of confidence in the forecasts dependent on location, lead time and indicator, as well as serving as a demonstration of a novel agricultural drought forecasting system driven by climate ensemble forecasts.

## 2 Data and models

### 2.1 SILO gridded climate data

SILO is a gridded dataset of climate data, mostly constructed from real measurements, that is used as the observational data. It is interpolated and infilled to give continuous coverage across Australia at 5 km resolution (Jeffrey et al., 2001), which makes it highly suitable for large simulation studies. In addition, it is already integrated with the AussieGRASS and APSIM simulation systems.

The complete set of target variables required for the biophysical and economic models are: minimum and maximum temperature (Tmin and Tmax), rainfall (Rain), incoming solar radiation (Radn), synthetic pan evaporation (Evap) and vapour pressure (Vapr). SILO data are aggregated in several ways for the purposes of climate forecast downscaling; more details are provided in section 3.1.

### 2.2 ACCESS-S2 climate model

Raw hindcasts of ACCESS-S2 (Wedd et al., 2022) are available for initialisation dates between 1981-01-01 and 2018-12-31. We make use of ensemble members generated on the 1st of each month as well as the preceding 8 days, which gives a raw ensemble comprising 27 members. Because the lag times are relatively short, the members are assumed to be exchangeable, which means they are treated as random draws from the same underlying distribution.

The climate variables used in the calculation of predictors are daily rainfall (pr), minimum temperature in 24 hours (tasmin),

maximum temperature in 24 hours (tasmax), net incoming shortwave solar radiation (rsds), specific humidity (huss), pressure (ps) and sea surface temperature (sst).



ACCESS-S2 raw data are on an approximate 80 x 60 km grid. Each run is initialised at midnight UTC and forecast variables are provided at a daily time step up to 215 days ahead. For the purposes of statistical forecast post-processing, we aggregate the forecasts to a monthly time step.

## 2.3 The AADI system

The AADI system is illustrated as a schematic in Figure 2. It comprises biophysical and agro-economic models driven by weather observations and forecasts. As this study focuses on the hindcasting performance, we refer to Hughes et al. (2024a) for details on the configuration of APSIM, AussieGRASS and *farmpredict*. However, for completeness, we give a summary of the critical details here.

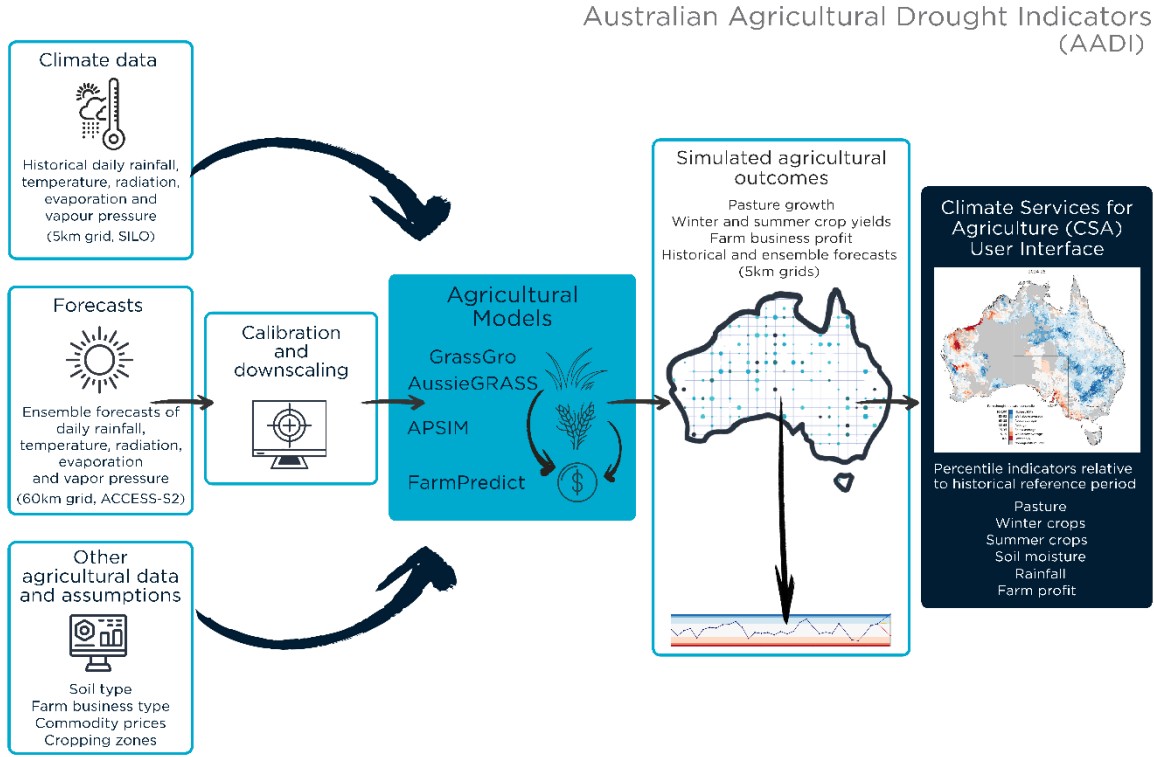


**Figure 2: Schematic of the Australian Agriculture Drought Indicators (AADI) system. Biophysical and agro-economic models are driven by daily weather observations or downscaled forecasts up to 18 months ahead on a 5 km grid across Australia. In the user-interface, drought indicators are calculated as percentiles against the historical distributions for display on user interfaces. In this study, the 51-member ensemble outputs from the agricultural models are verified on a reduced grid.**

APSIM simulates potential crop yield under different climatic conditions. For each hindcast year, APSIM is initialized with 15 years of historical weather data to establish equilibrium conditions, then run forward with SILO observations or ACCESS-S2 forecasts. Wheat simulations use cultivars optimised for yield in each grid cell, with specific management rules for sowing



and fertilization tailored to three regional zones. Sowing typically occurs between April and July, with nitrogen applied based on soil deficits and crop growth stages.

AussieGRASS, a pasture growth model operational for over 25 years, simulates pasture dynamics on a 5 km across Australia. Unlike other models that operate on a point-based system, AussieGRASS uses highly optimized code to run daily simulations across all grid cells, integrating tightly with SILO weather data. This model supports drought assessment and forecasting, providing insights into pasture availability under varying climatic conditions.

*Farmpredict* uses a statistical micro-simulation approach to model Australian broadacre farms, leveraging Australian

Agricultural and Grazing Industry Survey (AAGIS) data and machine learning (xgboost). It links farm characteristics, climate, and commodity prices to predict farm outputs and financial outcomes, including profit. Trained on 45,000 AAGIS observations from 1991–2022, *farmpredict* integrates geocoded farm data with SILO historical climate data to produce simulations of farm performance under different climatic and economic scenarios.

## Methods

**3.1 Climate forecast post-processing**

Climate forecast post-processing is required to reduce forecast biases, downscale to local features, extend forecast lead time and improve forecast reliability in ensemble spread. In a sense, post-processing the climate forecasts can be thought of as "pre-processing" for the purposes of agricultural simulations. However, we retain the term post-processing here as it is the usual term for such statistical methods.

We apply the Bayesian joint probability (BJP) modelling approach (Wang et al., 2019; Wang et al., 2009) coupled with an empirical downscaling technique (Potgieter et al., 2022; Schepen et al., 2020a, b). BJP is applied to calibrate climate forecasts on the coarse ACCESS-S2 grid. Then, the method of fragments (e.g. Westra et al., 2012) is applied to disaggregate forecasts to 5 km spatial resolution and to a daily time step, as described by Schepen et al. (2020a), who extended the MOF to disaggregate multiple variables simultaneously.

BJP applies transformed bivariate normal distributions to model the relationship between the predictor variables, which are taken as the means of the raw ACCESS-S2 ensembles, and observed variables, which are taken as the corresponding SILO observations. The transformation step accounts for non-normality, and non-continuous variables are handled through censoring in both the transformation and BJP models (Wang and Robertson, 2011; Wang et al., 2019). By conditioning the joint distribution on new values of the raw forecasts, a calibrated forecast distribution is obtained, from which a representative

ensemble can be sampled. We sample 51 ensemble members. An advantage of the BJP calibration, by virtue of being a model output statistics (MOS) approach, over simple bias corrections (such as linear scaling or quantile mapping) is that it returns the forecast to a climatology when the relationship between raw forecasts and observations is weak. In other words, it harnesses skill where available and returns a baseline forecast otherwise.



Raw versions of the target variables vapour pressure and evaporation are not directly available from ACCESS-S2; however,
we can approximate them from the available outputs using the following standard equations:

$$\text{vapr} = \frac{qP_s}{0.622 + 0.378q} \tag{1}$$

where $q$ is the specific humidity (kg/kg) and $P_s$ is the surface pressure (Pa).

$$\text{evap} = 0.0135(T_A + 17.78) * R_s * \left(\frac{238.8}{595.5 - 0.55T_A}\right) \tag{2}$$

where $T_A$ is the average surface temperature (°C) and $R_s$ is the surface solar radiation (MJ/m$^2$).

The main reason BJP is applied to monthly, coarse resolution forecasts is to calibrate forecasts at the scales of seasonal signals
and because calibration at a high spatial resolution and daily time steps is computationally expensive. However, efficient
spatial and temporal downscaling is then necessary. MOF is efficient but relies on having daily observational datasets at the
target resolution, which we have. In MOF, each BJP post-processed forecast ensemble member is matched to aggregated
historical observations through a nearest-neighbour search (Schepen et al., 2020a). The pattern of observations within the
month and at each high-resolution grid cell is used to simultaneously disaggregate the forecast, which generates sequences
with the correct intervariable, spatial and temporal correlations. Finally, the Schaake Shuffle (Clark et al., 2004) is applied to
every 5 km grid cell to link the ensemble members in grid cells across the continent to generate a forecast grid with correct
spatial, temporal and intervariable dependencies.

In addition to calibrating the ACCESS-S2 forecasts, the forecast ensembles are extended up to 18 months ahead to allow the
biophysical and economic models to be run over multiple financial years. Forecast augmentation occurs by introducing a "no-
input" BJP model for lead times beyond 6 months. The no-input models effectively model the marginal distributions of the
climate variables, and the outputs can therefore be seamlessly blended with the calibrated forecasts by disaggregating and
shuffling in the same way as the forecast ensemble members. In effect, the post-processed forecasts comprise 6 months of
forecasts and up to 12 months of model generated climatology.

### 3.2 Forecast verification

The AADI system produces raw outputs in the form of ensemble forecasts. It is therefore possible to assess the forecasts using
ensemble forecast verification methods, which is the primary way we assess forecast skill. The important aspects of forecasts
to be verified include accuracy, bias and reliability. We also assess forecast sharpness for additional context around these
metrics.

Verification of climate forecast post-processing is undertaken via a leave-one-year-out cross-validation framework. Whilst not
able to completely eliminate bias associated with training and evaluating over the same period (Risbey et al., 2021), cross-
validation is a means to obtain a realistic skill estimate when there is insufficient data for split-sample testing and remains
standard practice in seasonal forecasting.



Model based agricultural forecasts are verified via comparison with the pseudo-observations (i.e., ignoring model error). As such, the results offer an assessment of climate forecast skill in a specific agricultural context, but do not estimate the absolute skill of these models in forecasting on-the-ground agricultural outcomes (crop yields, pasture growth or farm profits). The related study of Hughes et al. (2024a) offers a detailed assessment of AADI performance against a range of observed ground-truth data, while skill assessments of the individual models have been published previously (for example, Hughes et al. (2022b) present leave-one-year-out cross validation results for *farmpredict*).

### 3.2.1 Ensemble scores

Percentage bias is calculated to assess systematic over- or under-prediction:

$$\text{Bias (\%)} = \frac{\sum_{t=1}^{T}(\overline{f_t} - o_t)}{\sum_{t=1}^{T} o_t} \times 100 \tag{3}$$

where $\overline{f_t}$ is the forecast ensemble mean for event $t$ and $o_t$ is the corresponding observation.

The continuous ranked probability score (CRPS) is a metric for evaluating the accuracy of an ensemble forecast considering the full distribution of the forecast:

$$\text{CRPS}_t = \int \left(F_t(x) - \mathbf{1}_{x \geq o_t}\right)^2 dx \tag{4}$$

Where $F_t$ is the cumulative distribution function representing the forecast ensemble for event $t$ and $\mathbf{1}_{x \geq o_t}$ is an indicator function representing the CDF of the observation, which is 1 for $x \geq o_t$ and 0 otherwise. We adopt the empirical calculation of CRPS as per Hersbach (2000).

The average CRPS over a set of forecasts is compared to a baseline or reference set of forecasts, and computed into a skill score that quantifies the percentage reduction in error:

$$\text{Skill score (\%)} = \left(1 - \frac{\overline{\text{CRPS}}}{\overline{\text{CRPS}}_{\text{Ref}}}\right) \times 100 \tag{5}$$

The skill score can reach a maximum of 100%, which indicates a perfect match between forecasts and observations. A skill score of 0% indicates that the forecasts and reference forecasts have similar overall performance. Negative skill scores are unbounded but indicate that the forecasts do not add value over knowledge of the historical (climatology) distribution.

Reliability is assessed through evaluation of the distribution of probability integral transforms (PITs):

$$\text{PIT}_t = F_t(o_t) \tag{6}$$

If the frequency of observations is consistent with forecast probabilities, then the PIT values over a set of events will be uniformly distributed (Diebold et al., 1998), which can be assessed using a standard uniform QQ plot (e.g. Wang and Robertson, 2011). The mean absolute deviation of the PITs from the 1:1 line can be summarised into an overall reliability metric:



$$\text{REL} = 1 - \frac{2}{M} \sum_{i=1}^{M} \left| \frac{i}{M+1} - \text{PIT}_{(i)} \right| \tag{7}$$

where $\text{PIT}_{(i)}$ is the $i$ th ranked PIT value in ascending order, and the score has been adjusted to range between 0 and 1, where 1 is perfect reliability is 0 is poor reliability.

In a properly calibrated ensemble forecasting system, forecast sharpness and forecast skill will be related. However, to aid in the interpretation of reliability wrt bias and forecast sharpness, we consider the relative width of the forecast ensemble interdecile range (IDR) over the climatological IDR as a measure of forecast ensemble spread (dispersion). If dispersion is 100%, then, on average, the forecast ensembles are the same width as the climatology. Typically, it is expected that the forecast ensemble spread narrows with increasing skill.

$$\text{Dispersion (\%)} = \frac{\sum_{t=1}^{T}(F_t^{-1}(0.9) - F_t^{-1}(0.1))}{\sum_{t=1}^{T}(F_{\text{clim},t}^{-1}(0.9) - F_{\text{clim},t}^{-1}(0.1))} \times 100 \tag{8}$$

where $F_t^{-1}$ and $F_{\text{clim},t}^{-1}$ are the inverse CDFs of the forecast and reference forecast, respectively.

### 3.2.2 Reference forecasts

A reference ensemble for assessing forecast skill is created by using the set of historical observations. In the leave-one-year-out cross-validation framework, a separate reference ensemble is assembled for each target year, by omitting the observation for the target year, and using the observations from all the other years to construct the ensemble. The reference ensemble is not influenced by the present year's antecedent conditions nor the climate forecast, and therefore sets the baseline for skill as knowledge of the historical spread of grain yields, pasture growth or farm profits. Whilst other baselines are possible, the use of these historical simulations as a reference is consistent with the construction of the indicators, which are expressed as percentiles of the historical simulations.

### 3.2.3 Spatial and temporal sampling

AADI is set up in real-time on a 5 km grid and runs every month. However, the computational and financial cost of running APSIM in all grid cells is prohibitive in the current technological environment. Therefore, for the wheat and sorghum simulations, we sample every 4<sup>th</sup> grid cell which, as can be seen in the results, gives considerable coverage. The AussieGRASS and *farmpredict* models retain full coverage. We also evaluate performance at only four times in the year, for forecasts beginning in April, July, October and January, which were selected to roughly align with agricultural industry decision points.



## 4 Results

### 4.1 Climate

We briefly present the skill of climate forecasts to analyse the contribution of climate to the overall skill of the drought indicators. The CRPS skill scores of the calibrated monthly climate forcings are summarised in

Figure 3. Each doughnut plot shows the median and interdecile range of skill for a climate variable at a lead time, for each forecast issue month. The percentiles are determined by pooling all grid cells and therefore indicate spatial coverage. At one month lead time, most of the climate forecasts have moderate skill with temperature, evaporation and vapour pressure being the mostly skilfully predicted variables. Radiation and rainfall have relatively low skill. Beyond one month lead time, low to moderate skill is evident in the 90[th] percentiles for temperature, evaporation and vapour pressure, indicating skill in some regions in the Austral Spring. Elsewhere, and at lead times beyond 2 months, skill is minimal. Slightly negative skill scores are evident, albeit expected in cross-validation.

Bias and reliability have also been evaluated and have been omitted for brevity given previous reporting and almost universal high reliability (PIT reliability scores > 0.8) and minimal bias (typically within ±5%). However, some locations show a moderate percentage bias (20%) for rainfall in the dry season, where low seasonal totals magnify relative errors and the prevalence of zeros make it more difficult to perfectly correct mean bias due to lower bound effects.

### 4.2 Farm profit

Maps of bias, reliability, dispersion and overall skill for the farm profit indicator are presented in Figure 4 for each of the forecast issue months. The farm profit indicator displays little or no bias (within ±5%) for forecasts issued in current financial year (FY) April, January and October. However, regions of positive bias exist in the central east for forecasts issued in current FY July and previous FY April (up to 20%). Skill increases markedly as the end of the financial year approaches, which is mainly due to an increasing proportion of observed data being integrated into the indicator. Even so, for July issued forecasts, which have a lead time of 12 months, the median CRPS skill score is 45% (23-64% interdecile range). This level of skill is quite high, when contrasted with the skill of climate forecasts, which typically have CRPS skill scores of less than 30% depending on the location and season, and typically not more than a few months ahead (see discussion in Section 5). We can infer that the high degree of skill in the farm profit indicator is primarily due to knowledge of the antecedent environmental and economic conditions rather than climate. Skill improves markedly as lead time decreases and for the current FY April forecasts, skill reaches approximately 80-90%, which suggests that predicted farm profits converge with many months lead time.

For each forecast issue month, the skill of farm profit predictions shows limited spatial variation, suggesting that forecast errors are relatively consistent across different climatic and agricultural zones. Reliability is high (> 0.8) for all forecast issue months and regions. High reliability indicates that the forecast probabilities are consistent with the frequency of outcomes and, therefore, the ensemble spread is typically unbiased and of appropriate spread. The relative width of the forecasts is also



measured through dispersion, showing that, compared to the historical distribution of profits, the forecast ensemble spreads tend to be about as wide as the historical distribution at long lead times. As forecast skill improves, the spreads narrow in concert with the increase in skill, suggesting that the economic forecasts remain well calibrated. Moreover, in cropping regions, dispersion is lower in January and April compared to (say) northern regions, which differentiates between completed

260  crops and pasture-dominant areas, consistent with farm profits in the winter cropping zone being highly dependent on April to October rainfall.

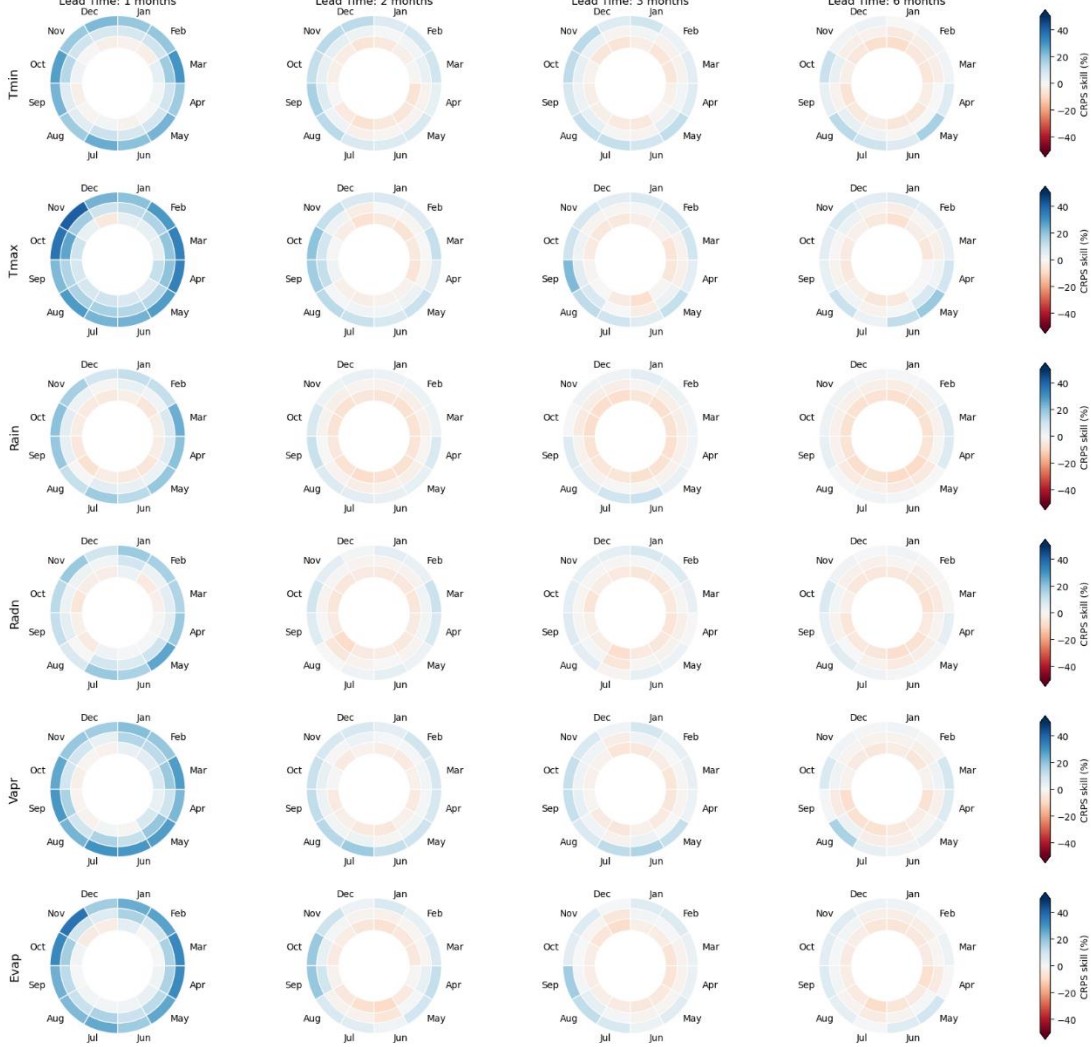

265  **Figure 3: Summary of CRPS skill scores of the climate forecasts over the climate hindcast period 1981-2018. The three rings depict the median (middle ring) and interdecile range (inner ring= 10th percentile; outer ring = 90th percentile). Each ring segment represents for a forecast issue month from January to December in a clockwise direction. Target climate variables are in rows and the lead time (months ahead) are in columns.**





**Figure 4: Forecast verification maps for the farm profit indicator calculated over the period 1990-2018. Columns are per verification metric: CRPS Skill Score, bias, reliability and dispersion. Rows are per forecast issue date: April in the previous financial year, then July, October, January and April of the current financial year.**





**Figure 5: Forecast verification maps for the APSIM potential wheat yield indicator calculated over the period 1990-2018. Columns are per verification metric: CRPS Skill Score, bias, reliability and dispersion. Rows are per forecast issue date: April in the previous financial year, then July, October and January of the current financial year.**





**Figure 6: Forecast verification maps for the APSIM potential sorghum yield indicator calculated over the period 1990-2018. Columns are per verification metric: CRPS Skill Score, bias, reliability and dispersion. Rows are per forecast issue date: July, October, January and April of the current financial year.**



### 4.3 Winter crops (wheat)

In Australia's warmer northern regions, wheat is typically sown from May to July and harvested from October to December,
while in cooler southern regions, sowing occurs from April to June with harvests from November to January, although there
can be variation outside these windows. The verification metrics for previous FY April and current FY July, October and
January forecast issue dates are mapped in Figure 5. January is the latest forecast issue date because the crops are harvested
by this time. Skill tends to be low for wheat around sowing time and, as to be expected, increases gradually as the season
progresses. For forecasts issued in July, the median CRPS skill score is 31% (14-56% IDR), which indicates moderate
association with historical simulations with 3-6 months lead time. By flowering and harvesting, skill is typically very high (80-
90%), indicating that the forecasts have converged and match the historical simulations well. Biases overall are small (within
±5%) except for the April issued forecasts, where positive biases appear in central Queensland, and negative biases appear in
southwest Western Australia. Reliability is high for forecasts issued in April, July and October. Reliability appears to decrease
for January issued forecasts, however, this is somewhat misleading, because the maturation of the crops means the ensemble
spreads are correspondingly very narrow (as indicated by dispersion). Consequently, it is possible for the observation to fall
marginally outside the ensemble, leading to poor probabilistic reliability, despite having a small absolute error.

### 4.4 Summer crops (Sorghum)

Sorghum crops are limited to regions in Queensland and NSW, often sown as an opportunistic crop in between regular wheat
planting. Sorghum has a wide window of sowing opportunity from September to January with the crop cycle taking 4-5 months
to complete.

The assessment of sorghum hindcasting skill is presented in Figure 6. Overall skill, in terms of the CRPS skill score, is only
evident after crop emergence, that is for forecast issue dates in January and April. The median skill for forecasts issued in
January is 31% (13-41% IDR) and median skill for forecasts issue in April is 51% (29-68% IDR), indicating low to moderate
forecasting skill. A mixture of positive and negative bias is evident across all forecast issue dates; the median absolute
magnitude of the percentage bias for the forecast issues months ranges from 6.2 to 7.7 %, suggesting a discrepancy has arisen
between the historical simulations (pseudo-observations) and the hindcast simulations (see section 5 for further discussion).
As with the wheat hindcast results, the sorghum simulations demonstrate high reliability, except for the April issued forecasts,
which reliability is artificially deflated by the very narrow spread of a grown crop.





## Pasture Growth (avg)

**Figure 7: Forecast verification maps for the AussieGRASS pasture growth indicator calculated over the period 1990-2018. Columns are per verification metric: CRPS Skill Score, bias, reliability and dispersion. Rows are per forecast issue date: April in the previous financial year, then July, October and January of the current financial year.**



## 4.5. Pasture growth

Pasture growth includes native, rainfall-driven pastures, improved pastures and cropping systems, making the interpretation of the simulations more complex. To make the interpretation of the pasture growth more similar to the other indicators, pasture growth over a financial year is the combination of historically simulated pasture growth up until the forecast issue date and the aggregation of pasture growth over the remaining months of the forecast year.

Compared to the cropping simulations, pasture simulations have larger biases, particularly in central and western Australia, and skill is overall lower (Figure 7). For July issued forecasts, the 12-month outlook has a median CRPS skill score of 13% (0-28% IDR) and the median CRPS skill for April issued forecasts, essentially a 3-month outlook on top of the 9 months accumulated growth is 75% (56-86% IDR). Reliability is typically high across all forecast issue months, with median reliability being approximately 0.9.

## 4.6 Historical events

To analyse how the retrospective forecasts evolve over time at a national scale, the percentiles of spatially averaged farm profit are plotted against the forecast issue month for each hindcast year, along with the final farm profit and, as a reference indicator, percentiles of 12-month observed rainfall deficits. Although the rainfall deficit percentiles do not directly correspond to final farm profit, their evolution can offer insights how such a lagging indicator, currently used for drought assessments, behaves in comparison with the forecast indicator.

We select a small number of the "worst" years for agriculture for detailed analysis based on the final farm profit percentile. The four worst years, from lowest to highest percentile are 2003, 2007, 2019 and 1995. For balance, we will also briefly consider the "good" years, all of which are visible in Figure 8.

Working in chronological order, the 1995 farm profits were affected by a drought built on back to back El Nino events in 1993-4 and 1994-5, with rainfall deficits most severe in the second half of 1994, before the drought broke in January 1995 (White et al., 1998; Lindesay, 2005). At 12 months lead time (July issue forecasts), the rainfall and farm profit indicators both indicate below average conditions, although the farm profit indicator provides earlier warning that the impacts are likely to be more severe. By 9 months lead time (October issue forecasts), the rainfall and farm profit indicators are both pointing to a severely impacted year.

The 2003 drought, the worst in terms of farm business outcomes, occurred in the middle of the Millennium drought. Long term rainfall deficits began dipping below normal in 2002 in response to low rainfall across most of the continent (Figure 8). The severity of the drought may have been somewhat surprising at the time given the weak to moderate level of the associated El Nino event (Mcphaden, 2004; Wang and Hendon, 2007), which highlights that a single climate driver cannot be relied upon as a signal of drought, and the antecedent conditions, such as existing hydrological drought, play a significant role. At 12-months lead time (July forecast issue), the farm profit and rainfall indicators both point to well below average conditions. As



the year progresses, both indicators continue to trend downwards, indicating that, in this case, the progression of meteorological drought was locked fairly in sync with the weaking outlook for farm profits.

After a brief reprieve, the 2007 farm profits were also impacted by the Millenium Drought and associated with the development

of a weak El Nino event (e.g. Su et al., 2018). With 12-months lead time, the farm profit indicator indicated a high likelihood of low profits. However, the 12-month rainfall indicator showed relatively normal rainfalls. In the following months, the rainfall indicator trends from slightly above average to slightly below average, whilst the farm profit indicator remains low. The reason for the rainfall indicator failing to drop low is that rainfalls in western Australia were above average, whilst meteorological drought conditions mainly intensified in the southeast.  Localised information is therefore critical consideration

in the use of indicators from a drought early warning system.

The final drought year analysed sits within the Tinderbox drought, which peaked in terms of farm profits impacted in 2019. It is only possible to analyse the first three forecasts for this event because the retrospective forecasts from ACCESS-S2 finish in 2018. We can see that in 2019, both the rainfall and profit indicators are indicating severe drought impacts with up to 12 months lead time. The 2018-19 drought rapidly transitioned to severe drought conditions following on from an exceptionally

wet period in 2016-17. In fact, the evolution of the drought indices through 2017 to 2019 shows that the farm profit indicator provided advanced indications of high farm profit with 12 months lead time in 2017 before leading the rainfall indicator into drought conditions in 2018.

Whilst the focus in this study in on drought, the performance of the drought indicators in normal and "good" years is also relevant, for example, in the context of false alarms. We can summarise that in 1994, 1997 and 2017, the farm profit indicator

provided advance information for high farm profits compared to the rainfall indicator, however, for 2012, rainfall remained high following the 2010-11 high rainfall, and the farm profit indicator was relatively late in identifying high profits, partially because the profit indicator "resets" at the beginning of the financial years whereas the rainfall indicator is continuous.





**Figure 8: Time series plots of the nationally averaged farm profit indicator (blue line) with the blue shading depicting the interdecile range of the ensemble. The x-axis depicts the forecast issue date and the y-axis is the percentile. The final farm profit for the financial year is plotted (green line) as well as the nationally averaged 12-month lagged rainfall percentile (orange line).**



## 5 Discussion

Climate forecast post-processing is an essential step to prepare raw climate forecast ensembles from a dynamical climate model for ingestion into the downstream biophysical and agro-economic models. Climate forecast skill is generally available for one month ahead across a set of climate variables (temperature, rainfall, radiation, evaporation and vapour pressure), with widespread CRPS skill scores up to 30% (Figure 3). Beyond the first month, limited skill is available in temperature, vapour pressure and evaporation forecasts, with little or no skill in rainfall and radiation forecasts. However, because calibrated climate

forecasts have little or no bias (except rainfall forecasts in dry regions can show moderate percentage bias) and high reliability in ensemble spread, the forecasts provide suitable forcings for the downstream models even where skill is limited.

Our results demonstrate much higher skill for the drought indicators compared to climate, which highlights the predictive importance of antecedent environmental and economic conditions in the downstream models. The AADI models are integrative, and capture not only rainfall, but important factors like antecedent soil moisture and prices over the preceding

months and years. The narrower ensemble spreads afforded by skilful forecasts provide drought policy analysts greater confidence in identifying areas requiring greater attention. Forecasts of all drought indicators show high reliability in ensemble spread, which supports their use for probabilistic decision making at an appropriate risk level.

Some moderate biases exist in farm profit, sorghum and pasture in central-eastern parts of Australia. Sometimes, such as with farm profit, these vanish shorter lead times. Such discrepancies between historical runs and hindcast simulations, especially

evidenced by bias in hindcasts after convergence is expected (e.g.Figure 6), highlight potential differences in configuration or input data when historical simulations and hindcasts were run on different computing infrastructure. In contrast, the wheat indicator is largely bias free across all lead times. Future work will focus on running all simulations on the same infrastructure to ensure consistency across both datasets to improve the reliability of performance evaluations.

Although month-to-month pasture results are not shown, they exhibit larger forecast biases compared to seasonal or annual

averages. In AussieGRASS, the parameterisation related to the soil water index, which controls plant growth onset and cessation, contributes to non-linear responses to rainfall. This sensitivity can amplify small biases in rainfall forecasts, leading to significant transient errors in modelled pasture growth. Ongoing refinement or recalibration of AussieGRASS parameters will aim to address this issue.

Although the farmpredict takes yields and pasture as inputs, biases observed in farm profit predictions at longer lead times,

particularly in areas just beyond the edge of the cropping zone, could be partially explained by the interpolation method used for input data (Hughes et al., 2024b). Farm input data interpolation was applied separately for rangelands and cropping zones, resulting in higher interpolation errors near their borders. Addressing this known issue will involve refining the interpolation method to reduce errors at the interface between these land-use types.



Training and testing models on historical data ensure that parameters reflect past climate conditions. While this is consistent
with current climate forecast evaluation periods, changing climate conditions could impact the skill of future predictions.
Recognising this, future research will explore strategies to maintain model relevance under evolving climatic scenarios.

Regarding improvements to the system that could enhance real-time skill, the real-time AADI climate forecast post-processing
system could benefit from dynamic training, incorporating the most recent forecast and observation data at each time step.
This allows the system to adapt to changing biases and be trained on more data to improve accuracy. In parallel, developing
emulators or error models to systematically correct residual biases between forecasts and "observed" simulations could
simultaneously improve forecast accuracy and address the problem of the crop models being computationally expensive.

A more efficient workflow will have the forecasts in the hands of drought analysts earlier, giving as much lead time as possible
to make drought policy conditions. Of course, there remains some discrepancies between the forecasts of drought indicators,
which occur in model space, and what occurs on the ground. The relationships between the AADI drought indicators and real-
world outcomes, like yield, and socio-economic data, are addressed by Hughes et al. (2024a). Nevertheless, studies like the
current one are needed to evaluate indicators, like pasture, for which there is no real observed data. As such, the AADI
indicators user-interface, used by the drought analysts, will indicate both 'forecast skill', i.e. ensemble forecast verification in
the model world, and 'indicator skill', which is cross-correlations with a multitude of real-world indicators.

## 6 Conclusion

This study has been a first step towards quantifying and understanding the performance of the Australian Agriculture Drought
Indicators as a forecasting system. The farm profit indicator has high CRPS skill when compared with the historical distribution
of simulated farm profits. Even at 12 months lead time, median farm profit skill is 45%, which exceeds climate forecast skill
and is attributable to the integration of antecedent environmental and economic conditions. Early season skill in wheat
predictions is moderate at about 30% (median), however, this increases to 79% and 90% in mid and late season, respectively.
In contrast, sorghum showed lower skill and small to moderate biases (5-15%) across the growing districts which warrant
further investigation. For pasture, long lead time CRPS skill (12 and 9 months) is typically below 20%, consistent with the
dependency on rainfall forecasts.

Historical event analyses show that, by considering the propagation of drought and the effect of commodity prices, the AADI
system has benefits over standard rainfall analysis for providing warning of drought effects in agriculture. Importantly, the
AADI system appears unbiased towards drought scenarios and tests well for "good" years, which is important for covering
drought and recovery. Future work will focus on eliminating biases in the system and improving overall skill, as well as
considering improvements to support use directly by industry.



## Acknowledgements

This work was funded by the Australian Government Department of Agriculture, Fisheries and Forestry. SILO gridded climate data was provided by the Queensland Government Department of Environment, Science and Innovation. ACCESS-S2 climate hindcasts were provided by the Australian Government Bureau of Meteorology. We thank Ross Searle who provided the national soil data used in the simulations and Geoffrey Brent and Andrew Turner who helped with the early *farmpredict* modelling efforts. Heidi Horan provided historical data from APSIM modelling and Yong Song contributed to the earlier
development of the statistical forecast post-processing methods.

## Code and data availability

Access to subsets of data and code can be arranged by contacting the corresponding author.

## Author contributions

Andrew Schepen led the hindcasting study and forecast post-processing. Andrew Bolt produced and analysed the forecast
verification metrics. Donald Gaydon and Zvi Hochman provided expertise on crop simulation modelling. John Carter and Dorine Bruget undertook AussieGRASS modelling. Neal Hughes, Peter Tan and Mihir Gupta contributed to *farmpredict* modelling. Chris Sharman and Peter Taylor developed operational modeling and data processes. Andrew Schepen wrote the manuscript with contributions from Neal Hughes, Zvi Hochman, Peter Taylor and Don Gaydon.

## Competing interests

The authors have no known competing interests.



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
