# Peer review of "Forecasting agricultural drought: the Australian Agricultural Drought Indicators"

_EGUsphere, 2024_

## Author Response (AR1)

Reviewer comments are in plain black text.

Our author responses are in blue italics.

Cuts from the manuscript are in green italics.

**Response to reviewer 1:**

This is my first review of the manuscript Forecasting Agricultural Drought: The Australian Agriculture Drought Indicators by Andrew Schepen et al. The paper is well-written, clear, and highly relevant to the journal. The topic is timely, as forecasting agricultural droughts is a critical area of research with significant implications.

However, I have a few moderate concerns, outlined below:

1. The study primarily focuses on the sensitivity of seasonal forecasting system performance in relation to crop production and farm profit, rather than assessing the system's ability to predict crop growth. This is because the observations used are not real but rather derived from the same system, forced by ground data. While I do not see this as a weakness, I believe the authors should clarify this distinction when presenting the paper and discussing the results. Additionally, if this study is more of a sensitivity analysis of forecast errors, it would be beneficial to explore the relationship between meteorological forecast errors and crop yield errors. This could provide valuable insights into conditions where the system may struggle to predict crop yield accurately.

RESPONSE: The reviewer is correct that our forecasts are evaluated against pseudoobservations, a concept which we introduce in lines 66-69. We completely agree about the importance of further comparing AADI predictions compare to real-world data. We have tweaked our discussion to make this clearer:

"Of course, there remain some discrepancies between the forecasts of drought indicators, which occur in model space, and what occurs on the ground. The relationships between the AADI drought indicators and real-world outcomes, like yield and socio-economic data, are addressed by Hughes et al. (2024a). Nevertheless, studies like the current one are needed to evaluate indicators, like pasture, for which there is no real observed data. As such, the AADI indicators user-interface, used by the drought analysts, will present the results of both studies and indicate 'forecast skill', i.e. ensemble forecast verification against pseudo-observations in the model world, and 'indicator skill', which is cross-correlations with a multitude of real-world indicators."

2. The dataset description is not sufficiently detailed. I suggest providing more information to improve clarity and transparency.

RESPONSE: We appreciate that the descriptions of the datasets are quite brief. Therefore, to provide more clarity on the datasets and how they are used, we have included a new table, identifying key datasets, their purpose, and spatial and temporal coverage in Table 1:

Table 1: Key datasets used for AADI forecast verification, description of their purpose, spatial resolution, and time periods.

| Dataset                                                       | Purpose                                                                          | Spatial resolution                       | Time period used |
|---------------------------------------------------------------|----------------------------------------------------------------------------------|------------------------------------------|------------------|
| ACCESS-S2 hindcasts                                           | Input - ensemble
forecasts to drive AADI
models                            | Native ~60 km grid
downscaled to 5 km | 1981-2018        |
| SILO climate grids                                            | Input - Forcing of baseline model runs; and downscaling of ACCESS-S2 forecasts   | 5 km grid                                | 1960-2018        |
| Australian Agricultural and
Grazing Industry Survey        | Input - training
farmpredict and
defining grid cell
characteristics     | Point data and regridded to 5km          | 1992-2022        |
| Soil type data, derived from the National Generic Soil Group. | Input - regional optimisation of APSIM                                           | Interpolated to 5km                      | Static           |
| Farm profit                                                   | Output – simulated
financial year profit
(Jun-Jul)                         | 5 km grid                                | 1990-2018        |
| Wheat potential yield                                         | Output – simulated
harvest yield (final
yield typically occurs
Sep-Jan) | 5 km grid limited to
wheat zones      | 1990-2018        |
| Sorghum potential yield                                       | Output – simulated
harvest yield (final
yield typically occurs
Mar-Jun) | 5 km grid limited to sorghum zones       | 1990-2018        |
| Pasture growth                                                | Output - average
growth over financial
year (Jun-Jul).                     | 5 km grid                                | 1990-2018        |

3. The **AADI system** should be described more thoroughly. For example, it is unclear whether **irrigation is considered** and how water limitations are accounted for. Additionally, what would be the impact of these factors? Given that some crops in Australia are irrigated, a discussion on this aspect would enhance the study's practical relevance.

RESPONSE: We agree that additional detail about the AADI system can be included. Currently, AADI produces water-limited yield, which represents the yield that can be

achieved using current best practices, technology and genetics for rainfed crops. We have updated to the introduction to read:

"Here, we connect climate forecasts from the Bureau of Meteorology's ACCESS-S2 seasonal model to farmpredict, crop models and a pasture model, considering only rainfed systems."

Furthermore, the section 2.3 is updated to provide more detail on APSIM and farmpredict:

"APSIM simulates potential crop yield under different climatic conditions. For each hindcast year, APSIM is initialized with 15 years of historical weather data to establish equilibrium conditions, then run forward with SILO observations or ACCESS-S2 forecasts. Wheat simulations use cultivars optimised for yield in each grid cell, with specific management rules for sowing and fertilization tailored to three regional zones. Sowing typically occurs between April and July, with nitrogen applied based on soil deficits and crop growth stages. Sorghum uses the 'Buster' variety with optimised density. Currently, AADI produces water-limited yield, which represents the yield that can be achieved using current best practices, technology and genetics for rainfed crops."

"Farmpredict uses a statistical micro-simulation approach to model Australian broadacre farms, leveraging Australian Agricultural and Grazing Industry Survey (AAGIS) data and machine learning (xgboost). It links farm characteristics, climate, and commodity prices to predict farm outputs and financial outcomes, including profit (July to June financial years). For example, farmpredict increases Australian fodder price and widens the Australian grain price basis (relative to global prices) when drought occurs. Trained on 45,000 AAGIS observations from 1991–2022, farmpredict integrates geocoded farm data with SILO historical climate data to produce simulations of farm performance under different climatic and economic scenarios."

We also refer to the companion Hughes et al study, also being revised for NHESS, which describes the system in more detail:

https://egusphere.copernicus.org/preprints/2024/egusphere-2024-3731/.

4. Has the system been tested **without post-processing**? If so, what is the impact? Including this analysis would provide valuable recommendations for the development of simpler systems in other regions.

RESPONSE: Omitting the climate forecast post-processing step is virtually certain to lead to poor performance due to climate model biases. Rather than updating our study to include simple post-processing methods, we lean on our previous studies and expertise in forecast post-processing to understand that failure to consider formal

calibration of ensemble forecasts will lead to unpredictable and often poor results. We have updated the first paragraph of our discussion to read:

"Climate forecast post-processing is an essential step to prepare raw climate forecast ensembles from a dynamical climate model for ingestion into the downstream biophysical and agro-economic models. Climate forecast skill is generally available for one month ahead across a set of climate variables (temperature, rainfall, radiation, evaporation and vapour pressure), with widespread CRPS skill scores up to 30% (Figure 3). Beyond the first month, limited skill is available in temperature, vapour pressure and evaporation forecasts, with little or no skill in rainfall and radiation forecasts. However, because calibrated climate forecasts have little or no bias (except rainfall forecasts in dry regions can show moderate percentage bias) and high reliability in ensemble spread, the forecasts provide suitable forcings for the downstream models even where skill is limited. While we have not tested raw forecasts or simple bias correction of climate forecasts, formal calibration ensures that forecast ensembles resemble Silo observations, which is vital for maintaining spatial, temporal and intervariable characteristics between forecasts and baselines."

Overall, the study is strong, but addressing these points would improve its clarity and impact.

**Response to reviewer 2:**

The manuscript titled "Forecasting Agricultural Drought: the Australian Agriculture Drought Indicators" presents a novel ensemble-based drought forecasting system that integrates post-processed seasonal climate predictions from the ACCESS-S2 model with biophysical and economic models. The authors address a topic of great significance in Australia, where forecasting drought is a primary concern for numerous stakeholders. The topic is presented in an excellent manner, and the manuscript includes a commendable literature review. The authors tackle a critical issue by evaluating forecast performance for four key drought-related indicators: wheat yield, sorghum yield, pasture growth, and farm profit over a historical hindcast period (1990 - 2018). The manuscript is methodologically robust, highly relevant, and well-written. However, some clarifications and refinements are needed. The following are suggestions for improvement:

1. The Introduction would benefit from an expanded discussion of major historical drought events in Australia, including impacts on agriculture. This sets the context more clearly for the need and relevance of AADI.

RESPONSE: We agree that additional scene-setting regarding major historical drought events will help justify the need and relevance of AADI. At present, we discuss historical

events in detail in section 4.6. We have updated the first paragraph of our introduction to read:

"Drought is a recurrent and significant challenge in Australia, which affects water resources, agriculture and ecosystems (Van Dijk et al., 2013; Devanand et al., 2024; Holgate et al., 2020; Lindesay, 2005). Two major droughts in recent decades are the Tinderbox Drought (2017–2020) and the Millenium Drought (2001–2009), which both had major impacts on industry and the environment. Even outside of drought periods, industries such as cropping and livestock are exposed to risks from high seasonal climate variability, long term declines in cool season rainfall (Mckay et al., 2023) and decadal monsoon variability (Heidemann et al., 2023). Historically, government responses to drought impacts in the agriculture sector have been informed by meteorological drought indicators such as rainfall deficits. However, a long history of practice has demonstrated that rainfall indicators are often poor flawed proxies for agricultural and economic drought impacts (Hughes et al., 2022a; Das et al., 2023; Stagge et al., 2015; Wang et al., 2022). In the absence of accurate assessments of agricultural impacts, government drought responses can be poorly directed, and overly reactive to media narratives (Rutledge-Prior and Beggs, 2021). Addressing these challenges requires not only monitoring of drought conditions but also forecasting of drought evolution, including both onset and recovery. (Das et al., 2023; Stagge et al., 2015; Wang et al., 2022)."

2. While the manuscript describes the datasets used, it would be helpful to include a consolidated table summarizing the key datasets, variables, spatial resolution, and time periods. This could be placed in the Methods section or as a supplement.

RESPONSE: We appreciate that the descriptions of the datasets are quite brief. Therefore, to provide more clarity on the datasets and how they are used, we have included a new table, identifying key datasets, their purpose, and spatial and temporal coverage in Table 1:

| Table 2: Key datasets used for                                                                                                                                                                                                                                                                                                                                                                                                                                                                                                                                                                                                                                                                                                                                                                                                                                                                                                                                                                                                                                                                                                                                                                                                                                                                                                                                                                                                                                                                                                                                                                                                                                                                                                                                                                                                                                                                                                                                                                                                                                                                                                 | or AADI forecast verification.        | description of their purpose. | spatial resolution, and time periods.     |
|--------------------------------------------------------------------------------------------------------------------------------------------------------------------------------------------------------------------------------------------------------------------------------------------------------------------------------------------------------------------------------------------------------------------------------------------------------------------------------------------------------------------------------------------------------------------------------------------------------------------------------------------------------------------------------------------------------------------------------------------------------------------------------------------------------------------------------------------------------------------------------------------------------------------------------------------------------------------------------------------------------------------------------------------------------------------------------------------------------------------------------------------------------------------------------------------------------------------------------------------------------------------------------------------------------------------------------------------------------------------------------------------------------------------------------------------------------------------------------------------------------------------------------------------------------------------------------------------------------------------------------------------------------------------------------------------------------------------------------------------------------------------------------------------------------------------------------------------------------------------------------------------------------------------------------------------------------------------------------------------------------------------------------------------------------------------------------------------------------------------------------|---------------------------------------|-------------------------------|-------------------------------------------|
| Thore is the set of th | · · · · · · · · · · · · · · · · · · · | description of their purpose, | sputtur resortation, until time periodist |

| Dataset                                                | Purpose                                                                        | Spatial resolution                       | Time period used |
|--------------------------------------------------------|--------------------------------------------------------------------------------|------------------------------------------|------------------|
| ACCESS-S2 hindcasts                                    | Input - ensemble
forecasts to drive AADI
models                          | Native ~60 km grid
downscaled to 5 km | 1981-2018        |
| SILO climate grids                                     | Input - Forcing of baseline model runs; and downscaling of ACCESS-S2 forecasts | 5 km grid                                | 1960-2018        |
| Australian Agricultural and
Grazing Industry Survey | Input - training
farmpredict and
defining grid cell
characteristics   | Point data and regridded to 5km          | 1992-2022        |

| Soil type data, derived from the National Generic Soil Group. | Input - regional optimisation of APSIM                                           | Interpolated to 5km                 | Static    |
|---------------------------------------------------------------|----------------------------------------------------------------------------------|-------------------------------------|-----------|
| Farm profit                                                   | Output – simulated
financial year profit
(Jun-Jul)                         | 5 km grid                           | 1990-2018 |
| Wheat potential yield                                         | Output – simulated
harvest yield (final
yield typically occurs
Sep-Jan) | 5 km grid limited to
wheat zones | 1990-2018 |
| Sorghum potential yield                                       | Output – simulated
harvest yield (final
yield typically occurs
Mar-Jun) | 5 km grid limited to sorghum zones  | 1990-2018 |
| Pasture growth                                                | Output - average
growth over financial
year (Jun-Jul).                     | 5 km grid                           | 1990-2018 |

 Was there any explicit quality control or filtering applied to the input data (e.g., rainfall, temperature) before simulation? If so, briefly describe this process.
 Otherwise, consider referencing previous studies that confirms the reliability of the datasets used.

RESPONSE: Regarding the observed climate data, the SILO gridded dataset of rainfall and temperature is a high-quality, managed dataset used widely in studies throughout Australia. We have updated section 2.1 to read:

"SILO is a gridded dataset of climate data, mostly constructed from real measurements, that is used as the observational data. It is interpolated and infilled to give continuous coverage across Australia at 5 km resolution (Jeffrey et al., 2001), which makes it highly suitable for large simulation studies. In addition, it is already integrated with the AussieGRASS and APSIM simulation systems. SILO is an operational product of the Queensland Government and is therefore continuously monitored and updated for quality."

4. Consider including a flowchart that visually summarizes the overall methodology (not only for the AADI system). A brief caption accompanying the figure would support reader comprehension.

RESPONSE: We have replaced the AADI schematic with visually simplified yet more complete description of the hindcasting workflow, including cross-validation, model simulations and forecast verification. See the new Figure 2:

Figure 1: Schematic of the workflows for 1) generating and verifying climate forecasts under leave one year out cross-validation and 2) subsequently generating and verifying profit, yield and pasture growth forecasts.

5. It would be better to clarify the spatial analysis methodology used when aggregating or averaging forecast data across Australia. Explain whether areaweighted averaging or other geostatistical techniques were applied to spatially aggregated time series.

RESPONSE: Averaging the forecasts across Australia for Figure 8 uses simple averaging. Section 4.6 has been updated to read:

"To analyse how the retrospective forecasts evolve over time at a national scale, the percentiles of spatially averaged farm profit are plotted against the forecast issue month for each hindcast year, along with the final farm profit and, as a reference indicator, percentiles of 12-month observed rainfall deficits. All grid cells are weighted equally in the averaging. Although the rainfall deficit percentiles do not directly correspond to final farm profit, their evolution can offer insights how such a lagging indicator, currently used for drought assessments, behaves in comparison with the forecast indicator."

6. The study focuses on model-based skill evaluation using pseudo-observations. Please address the rationale for this choice in more detail. How do the authors justify relying on model-to model comparison without validating forecasts against real ground-truth data?

RESPONSE: The reviewer is correct that our forecasts are evaluated against pseudoobservations, a concept which we introduce in lines 66-69. We completely agree about the importance of further comparing AADI predictions compare to real-world data. We have tweaked our discussion to make this clearer:

"Of course, there remain some discrepancies between the forecasts of drought indicators, which occur in model space, and what occurs on the ground. The relationships between the AADI drought indicators and real-world outcomes, like yield and socio-economic data, are addressed by Hughes et al. (2024a). Nevertheless, studies like the current one are needed to evaluate indicators, like pasture, for which there is no real observed data. As such, the AADI indicators user-interface, used by the drought analysts, will present the results of both studies and indicate 'forecast skill', i.e. ensemble forecast verification against pseudo-observations in the model world, and 'indicator skill', which is cross-correlations with a multitude of real-world indicators."

7. Explain the rationale for using ACCESS-S2 exclusively. If alternatives (e.g., ECMWF, SEAS5, NMME) were considered, briefly note why ACCESS-S2 was selected.

RESPONSE: One of the reasons for selecting ACCESS-S2 was the availability of realtime forecasts at the commencement of the project. Section 2.2 now reads:

"Climate forecasts are sourced from ACCESS-S2, selected as the model for real-time forecasting due to daily updates supporting timely forecast release. For this retrospective testing, raw hindcasts of ACCESS-S2 (Wedd et al., 2022) are available for initialisation dates between 1981-01-01 and 2018-12-31."

8. In the Discussion section, please consider addressing the uncertainties associated with the input datasets and how these may affect the reliability of the forecast outputs.

RESPONSE: We have provided some discussion around input errors and potential sources of errors, however we have expanded this discussion, which now reads:

"Some moderate biases exist in farm profit, sorghum and pasture in central-eastern parts of Australia. Sometimes, such as with farm profit, these vanish shorter lead times. Such discrepancies between historical runs and hindcast simulations, especially after convergence is expected (e.g. Figure 6), highlight potential differences in configuration or input data when historical simulations and hindcasts were run on different computing infrastructure. In contrast, the wheat indicator is largely bias free across all lead times. Future work will focus on running all simulations on the same infrastructure to ensure consistency across both datasets to improve the reliability of the predictions and minimise bias.

Although month-to-month pasture results are not shown, they exhibit larger forecast biases compared to seasonal or annual averages. In AussieGRASS, the parameterisation related to the soil water index, which controls plant growth onset and cessation, contributes to non-linear responses to rainfall. This sensitivity can amplify small biases in rainfall forecasts, leading to significant transient errors in modelled pasture growth. Ongoing refinement or recalibration of AussieGRASS parameters will aim to address this issue.

Although farmpredict takes yields and pasture as inputs, biases observed in farm profit predictions at longer lead times, particularly in areas just beyond the edge of the cropping zone, could be partially explained by the interpolation method used for input data (Hughes et al., 2024b). Farm input data interpolation was applied separately for rangelands and cropping zones, resulting in higher interpolation errors near their borders. Addressing this known issue will involve refining the interpolation method to reduce errors at the interface between these land-use types. Other errors may exist in input data such as SILO, however, these data errors are accounted for in calibrating climate forecasts to the SILO target. Soil type data is optimised on a grid cell basis and therefore may deviate from very local conditions at a paddock scale, and therefore it is not recommended to interpret the forecasts at a finer scale."

**Response to Reviewer 3**

The paper addresses an important and timely topic of drought forecasting for agricultural enterprises, going beyond rainfall indices. Its main objective is to assess the forecasting performance of multiple drought indictors within the AADI system. The results show that farm profit indicator has a high forecasting skill in comparison to rainfall indices. Assessment of historical events shows the usefulness of this indicator in predicting drought impacts. Overall, the manuscript is well written, but some clarifications and improvements will be useful for the readers.

**General Comments**

1. There appears to be some inconsistency in abstract and introduction regarding what is being assessed. From the abstract, I thought the paper will use commodity prices (along with crop growth) to forecast occurrence of droughts. But then in lines 52-53, it appears the goal is to forecast farm enterprises financial performance i.e. impacts of droughts using farmpredict model. And then again in line 70, there is mention of using farm profit as drought indicator. It would be helpful if the introduction consistently explains whether the paper aims to forecast drought via farm profit or to evaluate farm profit for drought impact assessment. Additionally, the introduction will also benefit with an explicit statement of its main aims, such as: "The aim of this study is..."

RESPONSE: We agree that there are some inconsistencies in the wording which can be improved. AADI forecasts seasonal and annual outlooks of agriculturally relevant drought indicators, of which farm profit is one indicator in addition to crop yield and pasture growth. We have modified the first part of the abstract to make it clearer that we are building a forecasting system rather than an attribution system:

"Drought is a recurrent and significant driver of stress on agricultural enterprises in Australia. Historically, rainfall indices have been used to identify drought and inform government responses. However, rainfall indicators may not fully reflect agricultural or economic drought conditions and are a lagging indicator. To address these shortcomings, AADI (Australian Agriculture Drought Indicators) was recently developed to monitor and forecast drought for upcoming seasons using biophysical and agroeconomic models, including crop yields, pasture growth, and farm profit at ~ 5 km2 resolution. Here, we evaluate the skill of drought indicator forecasts driven by the ACCESS-S2 dynamical global climate model over a hindcast period from 1990–2018."

2. It would help to specify the months, fiscal year boundaries, or assumptions that go into your farm profit calculations, so Section 4.2 can be understood without needing to reference other works.

RESPONSE: We agree that the results should be interpretable without heavily relying on other works. We have now added Table 1, which describes the relevant target months and/or financial year boundaries for each target variable. We have also expanded the description of farmpredict in section 2.3 (see response to #5 below). However, we are revising two papers simultaneously for publication in NHESS, so we need to strike a balance with the level of detail that can be found in https://egusphere.copernicus.org/preprints/2024/egusphere-2024-3731/.

Table 3: Key datasets used for AADI forecast verification, description of their purpose, spatial resolution, and time periods.

| Dataset                                                       | Purpose                                                                                    | Spatial resolution                       | Time period used |
|---------------------------------------------------------------|--------------------------------------------------------------------------------------------|------------------------------------------|------------------|
| ACCESS-S2 hindcasts                                           | Input - ensemble
forecasts to drive AADI
models                                      | Native ~60 km grid
downscaled to 5 km | 1981-2018        |
| SILO climate grids                                            | Input - Forcing of
baseline model runs;
and
downscaling of
ACCESS-S2 forecasts | 5 km grid                                | 1960-2018        |
| Australian Agricultural and
Grazing Industry Survey        | Input - training farmpredict and defining grid cell characteristics                        | Point data and regridded to 5km          | 1992-2022        |
| Soil type data, derived from the National Generic Soil Group. | Input - regional optimisation of APSIM                                                     | Interpolated to 5km                      | Static           |

| Farm profit             | Output – simulated
financial year profit
(Jun-Jul)                         | 5 km grid                          | 1990-2018 |
|-------------------------|----------------------------------------------------------------------------------|------------------------------------|-----------|
| Wheat potential yield   | Output – simulated
harvest yield (final
yield typically occurs
Sep-Jan) | 5 km grid limited to wheat zones   | 1990-2018 |
| Sorghum potential yield | Output – simulated
harvest yield (final
yield typically occurs
Mar-Jun) | 5 km grid limited to sorghum zones | 1990-2018 |
| Pasture growth          | Output - average
growth over financial
year (Jun-Jul).                     | 5 km grid                          | 1990-2018 |

3. Over the hindcast period, the farm profit indicator is shown to have a good predictive skill as presented in section 4.6. However, the performance is assessed for historically declared droughts. It doesn't become immediately clear to me how such prediction will be used in real-time forecasting to attribute changes in farm profits to droughts. I assume from the information provided in section 5 about inclusion of factors like soil moisture (Lines 383-385) within the AADI system, however, such information is not presented in the methods section and should be included briefly despite the reference to Hughes et al. (2024a).

RESPONSE: The purpose is not to strictly attribute changes in farm profit to drought, but to provide early warning of where agricultural enterprises may be impacted by the climate, antecedent conditions, economics or some combination of these. In this sense, AADI has broader application in seasonal outlooks although drought is the central focus.

4. From the manuscript, it doesn't become completely clear to me how the paper addresses the stated shortcoming of drought propagation (line 14-15) and drought evolution (Line 36-37) in the drought forecasts.

RESPONSE: The statements about drought propagation and evolution are more criticisms of standard practice to use lagged rainfall indicators as a measure of drought. AADI is a fully integrated system that takes weather forecasts and integrates them through a suite of agricultural and economic models, to predict future conditions that may include the onset or conclusion of drought, through an agricultural lens. For example, it is quite possible that high soil moisture stores sustain a high yielding crop, despite low rainfall. However, as noted in our response above, we have modified the abstract to no longer mention propagation. We have also changed the first paragraph of the introduction, which no longer mentions drought evolution and adds more context on historical droughts:

"Drought is a recurrent and significant challenge in Australia, which affects water resources, agriculture and ecosystems (Van Dijk et al., 2013; Devanand et al., 2024; Holgate et al., 2020; Lindesay, 2005). Two major droughts in recent decades are the Tinderbox Drought (2017–2020) and the Millenium Drought (2001–2009), which both had major impacts on industry and the environment. Even outside of drought periods, industries such as cropping and livestock are exposed to risks from high seasonal climate variability, long term declines in cool season rainfall (Mckay et al., 2023) and/or decadal monsoon variability (Heidemann et al., 2023). Historically, government responses to drought impacts in the agriculture sector have been informed by meteorological drought indicators such as rainfall deficits. However, a long history of practice has demonstrated that rainfall indicators are often flawed proxies for agricultural and economic drought impacts (Hughes et al., 2022a; Das et al., 2023; Stagge et al., 2015; Wang et al., 2022). In the absence of accurate assessments of agricultural impacts, government drought responses can be poorly directed, and overly reactive to media narratives (Rutledge-Prior and Beggs, 2021). Addressing these challenges requires not only monitoring of drought conditions but also forecasting of drought onset and recovery. (Das et al., 2023; Stagge et al., 2015; Wang et al., 2022)."

5. Drought can raise or lower commodity prices. Is this aspect captured in your profit-based indicator, or are prices assumed exogenous? Please clarify.

**RESPONSE: Yes. Our section 2.3 now reads:**

"Farmpredict uses a statistical micro-simulation approach to model Australian broadacre farms, leveraging Australian Agricultural and Grazing Industry Survey (AAGIS) data and machine learning (xgboost). It links farm characteristics, climate, and commodity prices to predict farm outputs and financial outcomes, including profit (July to June financial years). For example, farmpredict increases Australian fodder price and widens the Australian grain price basis (relative to global prices) when drought occurs. Trained on 45,000 AAGIS observations from 1991–2022, farmpredict integrates geocoded farm data with SILO historical climate data to produce simulations of farm performance under different climatic and economic scenarios."

6. The Discussion should reference related work on drought forecasting indicators in agriculture to help place your results in wider context.

RESPONSE: Our introduction sets the scene for our work in the context of agricultural drought indicators, particularly in Australia. However, to tie up the global context in our discussion, we have added the following paragraph:

"In the global context, Oyarzabal et al. (2025) reviewed drought forecasting, albeit with a focus on machine learning. It was found that the vast majority of drought prediction

studies focus on meteorological drought and rainfall prediction, with relatively small focus on agricultural drought (13%). Moreover, most studies focussed on drought prediction indices such as SPI and SPEI. AADI has demonstrated, that in data-rich environments, it is feasible to develop a system of drought prediction that covers meteorological, agricultural and economic drought using hybrid approaches combining machine learning and process-based methods. However, we do see gains in developing ML based emulators and error models improve forecast accuracy relative to ground truth data and to overcome the problem of the crop models being computationally expensive, and which opens up greater opportunity to expand forecasts into data sparse regions."

**Specific comments**

1. Lines 62-63: "...policy planners might be interested in outlook for winter and summer crops..." Do you mean total production and not only yields? Please clarify.

RESPONSE: We clarified this refers to potential yield in our context, although interest in both is possible: "Indeed, in addition to farm profit outlooks, drought analysts and policy planners can also be interested in the potential yield outlooks for winter and summer crops, or pasture availability for livestock."

Line 72-73: Please give examples for threshold and categorical forecasts.

RESPONSE: We have given terciles as an example. "Often, drought system performance is evaluated using threshold or categorical forecasts (e.g. bottom tercile)."

2. Line 95: It would be useful for the readers to start with a brief introduction to ACCESS-S2 climate model.

RESPONSE: We have added a brief description: "ACCESS-S2, a global dynamical climate model from the Australian Bureau of Meteorology that provides forecast ensembles up to 6 months ahead"

3. Line 138: "MOF" used without definition

RESPONSE: Thank you, put MOF after the definition of the previous line: "Then, the method of fragments (MOF)..."

4. Line 231-233: "..... indicating skill in some regions in the Austral Spring." It is not clear how you reached this result. Either mention in text that this is not shown or add reference figure.

RESPONSE: As per line 229, the percentiles are calculated on spatial pooling, and therefore the conclusion about skill occurring in some regions follows naturally.

**5. Line 245: IDR instead of interdecile range**

RESPONSE: Corrected, thank you.